# Herpes Simplex Virus Re-Activation in Patients with SARS-CoV-2 Pneumonia: A Prospective, Observational Study

**DOI:** 10.3390/microorganisms9091896

**Published:** 2021-09-07

**Authors:** Erica Franceschini, Alessandro Cozzi-Lepri, Antonella Santoro, Erica Bacca, Guido Lancellotti, Marianna Menozzi, William Gennari, Marianna Meschiari, Andrea Bedini, Gabriella Orlando, Cinzia Puzzolante, Margherita Digaetano, Jovana Milic, Mauro Codeluppi, Monica Pecorari, Federica Carli, Gianluca Cuomo, Gaetano Alfano, Luca Corradi, Roberto Tonelli, Nicola De Maria, Stefano Busani, Emanuela Biagioni, Irene Coloretti, Giovanni Guaraldi, Mario Sarti, Mario Luppi, Enrico Clini, Massimo Girardis, Inge C. Gyssens, Cristina Mussini

**Affiliations:** 1Infectious Diseases Unit, Azienda Ospedaliero-Universitaria Policlinico, 41124 Modena, Italy; antonella.santoro7@gmail.com (A.S.); marymenozzi@gmail.com (M.M.); mariannameschiari1209@gmail.com (M.M.); andreabedini@yahoo.com (A.B.); gabriella.orlando7@virgilio.it (G.O.); cinzia.puzzolante@gmail.com (C.P.); margheritadigaetano@alice.it (M.D.); fedecarli@gmail.com (F.C.); gian.cuomo@gmail.com (G.C.); corradi.luca59@gmail.com (L.C.); 2Centre for Clinical Research, Epidemiology, Modelling and Evaluation, Institute for Global Health, University College London, London NW3 2PF, UK; a.cozzi-lepri@ucl.ac.uk; 3Department of Infectious Diseases, University of Modena and Reggio Emilia, 41124 Modena, Italy; erica.bacca@gmail.com (E.B.); g.lancellotti95@gmail.com (G.L.); jovana.milic@gmail.com (J.M.); giovanni.guaraldi@unimore.it (G.G.); 4Microbiology and Virology Unit, Azienda Ospedaliero-Universitaria Policlinico, 41124 Modena, Italy; gennari.william@aou.mo.it (W.G.); pecorari.monica@policlinico.mo.it (M.P.); sarti.mario@aou.mo.it (M.S.); 5Infectious Diseases Unit, G. da Saliceto Hospital, 29121 Piacenza, Italy; fluppi45@gmail.com; 6Nephrology, Dialysis and Transplant Unit, Azienda Ospedaliero-Universitaria Policlinico, 41124 Modena, Italy; gaetanoalfano85@gmail.com; 7Respiratory Diseases Unit, University of Modena and Reggio Emilia, 41124 Modena, Italy; roberto.tonelli@me.com (R.T.); enrico.clini@unimore.it (E.C.); 8Gastroenterology Unit, Azienda Ospedaliero-Universitaria Policlinico, 41124 Modena, Italy; nicolademaria@alice.it; 9Intensive Care Unit, Department of Anaesthesia, Azienda Ospedaliero-Universitaria Policlinico, 41124 Modena, Italy; stefano.busani@unimore.it (S.B.); emanuela.biagioni@gmail.com (E.B.); irene.coloretti@gmail.com (I.C.); girardis.massimo@unimore.it (M.G.); 10Hematology Unit, University of Modena and Reggio Emilia, 41124 Modena, Italy; mario.luppi@unimore.it; 11Radboud Center for Infectious Diseases, Department of Internal Medicine, Radboud University Medical Center, 6525 GA Nijmegen, The Netherlands; Inge.Gyssens@radboudumc.nl; 12Faculty of Medicine and Life Sciences, Hasselt University, 3500 Hasselt, Belgium

**Keywords:** SARS-Cov-2, *Herpesviridae*, steroids, tocilizumab, re-activation

## Abstract

Background: Herpes simplex 1 co-infections in patients with COVID-19 are considered relatively uncommon; some reports on re-activations in patients in intensive-care units were published. The aim of the study was to analyze herpetic re-activations and their clinical manifestations in hospitalized COVID-19 patients, performing HSV-1 PCR on plasma twice a week. Methods: we conducted a prospective, observational, single-center study involving 70 consecutive patients with severe/critical SARS-CoV-2 pneumonia tested for HSV-1 hospitalized at Azienda Ospedaliero-Universitaria of Modena. Results: of these 70 patients, 21 (30.0%) showed detectable viremia and 13 (62%) had clinically relevant manifestations of HSV-1 infection corresponding to 15 events (4 pneumonia, 5 herpes labialis, 3 gingivostomatitis, one encephalitis and two hepatitis). HSV-1 positive patients were more frequently treated with steroids than HSV-1 negative patients (76.2% vs. 49.0%, *p* = 0.036) and more often underwent mechanical ventilation (IMV) (57.1% vs. 22.4%, *p* = 0.005). In the unadjusted logistic regression analysis, steroid treatment, IMV, and higher LDH were significantly associated with an increased risk of HSV1 re-activation (odds ratio 3.33, 4.61, and 16.9, respectively). The association with the use of steroids was even stronger after controlling for previous use of both tocilizumab and IMV (OR = 5.13, 95% CI:1.36–19.32, *p* = 0.016). The effect size was larger when restricting to participants who were treated with high doses of steroids while there was no evidence to support an association with the use of tocilizumab Conclusions: our study shows a high incidence of HSV-1 re-activation both virologically and clinically in patients with SARS-CoV-2 severe pneumonia, especially in those treated with steroids.

## 1. Introduction

Severe acute respiratory syndrome coronavirus 2 (SARS-CoV-2) pneumonia is the most relevant clinical presentation of COVID-19. SARS-CoV-2 infection may trigger an uncontrolled systemic inflammatory response (the so-called “cytokine storm”) that is associated with acute respiratory distress syndrome, lung fibrosis, and long-term adverse outcomes [1,2]. This inflammatory phase is also characterized by an immunosuppression state and lymphopenia that is associated with poor outcomes [3,4]. Until now, a few drugs have been approved to treat patients with SARS-CoV-2 pneumonia. In particular, despite initial controversy, steroids were proved to reduce mortality in a large randomized controlled trial [5]. Moreover, other studies suggested a benefit of either antivirals as remdesivir and/or immunomodulatory drugs as baricitinib and IL-6 antagonists (tocilizumab and sarilumab) [6,7,8,9,10,11,12]. The efficacy of both antivirals and immunomodulatory drugs depends on the timing of administration. While remdesevir is useful during viral replication, steroids and immunomodulatory drugs are effective against the cytokine storm and can be harmful if used in the first days of SARS-CoV-2 infection [5].

The risk of super-infection and viral re-activation is not well known and may be caused by the interplay between SARS-CoV-2 infection, steroid use, and immunomodulatory drugs.

Recent literature suggests that *Herpesviridae* re-activations are frequent in intensive care unit (ICU) patients with COVID-19. In particular Herpes simplex virus (HSV) and cytomegalovirus (CMV) re-activations have been reported at higher rates in these patients than those described in previous studies in critically ill patients without SARS-CoV-2 infection [13]. The authors suggested lymphopenia as the probable cause of the increased incidence of *Herpesviridae* re-activation, without considering the use of steroids or tocilizumab [13,14]. Regarding patients in non-ICU wards, only case reports on skin, oral, and ocular Herpesviridae clinical manifestations in patients with COVID-19 were published [15,16,17,18].

The aim of this analysis was to evaluate HSV-1 re-activation by means of plasma polymerase chain reaction (PCR) and its clinical presentation in hospitalized patients with severe/critical SARS-CoV-2 pneumonia.

## 2. Methods

We conducted a prospective, observational, single-center study analyzing HSV-1 re-activation in COVID-19 patients admitted to the Azienda Ospedaliero-Universitaria of Modena, Italy. Between 8 April and 31 May 2020, we performed plasma qualitative PCR for HSV-1 twice a week in adult COVID-19 patients admitted to hospital. In patients who tested positive for qualitative HSV-1 PCR, quantitative HSV-1 PCR was performed (see Appendix A for technical specifications). The date of 8 April was chosen after 2 fatal cases of HSV-1 fulminant hepatitis occurred [19] It was decided to test all admitted COVID-19 patients in order to quantify the risk of herpetic re-activation. At a later time, in a subset of the study population, we retrospectively performed HSV-1 PCR on plasma that was stored at −80 °C) from patients at baseline, i.e., before immunosuppressive treatment, in order to determine if re-activation was already present at hospital admission. Each patient was followed until discharge or death or up to 15 June 2020.

In the same period, we also evaluated cytomegalovirus (CMV) re-activations with blood CMV-DNA; we consider as “high viremia” CMV-DNA > 10,000 UI/mL (see Appendix A for technical specifications).

The Institutional Ethics Committee of Area Vasta Emilia Nord (CE AVEN) approved the study (approval number 396/2020/OSS/AOUMO—CoV-2 MO-Study) on 5 May 2020. Due to the observational nature of the study, written informed consent was not required.

All patients presented respiratory symptoms and were diagnosed with SARS-CoV2 by real-time PCR performed on oropharyngeal swab specimens.

Data were collected from electronic medical records, including demographics, biomarkers of inflammation and coagulation, immunosuppressive drugs, PCR for HSV-1 on different patient specimens (plasma, bronchoalveolar lavage (BAL), cerebrospinal fluid, biopsy), clinical manifestations compatible with HSV-1 re-activation, antiviral prophylaxis or therapy, and outcomes (use of invasive mechanical ventilation (IMV), and death).

All microbiological samples were analyzed in the local Microbiology and Virology Laboratory. Technical specifications of HSV-1 PCR and CMV-DNA are described in Appendix A.

### 2.1. Definitions

Severe COVID-19 infection: individuals who have SpO_2_ < 94% on room air at sea level, a ratio of arterial partial pressure of oxygen to fraction of inspired oxygen (PaO_2_/FiO_2_) < 300 mm Hg, respiratory frequency > 30 breaths/min, or lung infiltrates > 50%.

Critical COVID-19 infection: individuals who have respiratory failure, septic shock, and/or multiple organ dysfunction.

### 2.2. HSV-1 Re-Activation

HSV-1 re-activation was diagnosed either if target was above the limit of detection (qualitative result) on plasma or >10,000 copies/mL in BAL quantitative assay [20].

### 2.3. Standard of Care (SOC)

All patients received SOC treatment at hospital admission according to regional COVID-19 guidelines of Emilia Romagna [21], and data on treatment of COVID-19 was updated in April 2020. SOC treatment included oxygen supply to target SaO_2_ reaching at least 90%, hydroxychloroquine (400 mg twice on day 1, followed by 200 mg twice per day on days 2–5, adjusted for estimated creatinine clearance), and low molecular weight heparin for prophylaxis of deep vein thrombosis according to bodyweight and renal function.

### 2.4. Tocilizumab Treatment

Patients were considered eligible for tocilizumab treatment if they showed a PaO_2_/FiO_2_ ratio < 200 mm Hg or SaO_2_ < 93% and a PaO_2_/FiO_2_ ratio < 300 mm Hg on room air associated with a >30% decrease in PaO_2_/FiO_2_ ratio in the previous 24 h during hospitalization.

Tocilizumab was administered by intravenous or subcutaneous route depending on the availability of specific formulation. Intravenous tocilizumab was administered at 8 mg/kg bodyweight (up to a maximum of 800 mg) twice, 12 h apart [7,22]. The subcutaneous formulation was used when there was a shortage of the intravenous formulation, at a dose of 162 mg administered in 2 simultaneous doses (i.e., 324 mg in total). In the analysis, exposure to tocilizumab was fitted as a binary variable (exposed vs. not exposed, regardless of the formulation).

### 2.5. Steroid Treatment

Steroids were not routinely administered during the study period (April–May 2020) outside the ICU; only patients with concomitant COPD received low-dose steroids (methylprednisolone 20 or 40 mg/day). In patients admitted to ICU steroids were administered for the prevention of pulmonary fibrosis and subsequently defined as ARDS treatment. Methylprednisolone was administered intravenously with an initial bolus of 0.5 mg/kg followed by administration of 0.5 mg/kg 4 times daily for 7 days, 0.5 mg/kg 3 times daily from day 8 to day 10, 0.5 mg/kg 2 times daily at days 11 and 12 and 0.5 mg/kg once daily at days 13 and 14. Of note, in case of failed response to tocilizumab, a methylprednisolone bolus at 1 g/d intravenously for 3 consecutive days was administered [23,24]. In the analysis, exposure to steroids was fitted in 3 separate models as: (i) binary variable (use vs. no use), (ii) two binary variables comparing low dosage and high dose (bolus or ARDS) vs. no use, and (iii) per day of any use.

### 2.6. Statistical Analysis

The main characteristics of the participants at hospital admission, comorbidities, signs and symptoms, treatment received, and median biomarker levels were compared by HSV-1 status using Chi-square or Mann–Whitney U test, as appropriate.

The analysis focused on the risk of HSV-1 re-activation associated with the use of tocilizumab and with the use of steroids in separate models.

The association with dose and duration of therapy with steroids was also evaluated in separate models. Unadjusted and adjusted logistic regression models with HSV-1 re-activation as the binary response variable were fitted.

A different set of potential confounding factors was hypothesized in the 2 models (Appendix A, Appendix A). For the tocilizumab model, baseline PaO_2_/FiO_2_ ratio was the only potential confounder, and steroid use was instead considered a mediator (Appendix A, Appendix A). Indeed, during the first wave of the pandemic, a key trigger for steroid initiation in our clinic was the previous failure of tocilizumab treatment. In contrast, when use of steroids was chosen as the main exposure of interest, the directed acyclic graph (DAG) suggested that controlling for both previous tocilizumab use and of IMV was sufficient to block all the backdoor confounding pathways (Appendix A, Appendix A).

We also investigated whether the impact of steroids on the risk of HSV re-activation might vary in people who were concomitantly treated with tocilizumab or not by formally testing for the interaction between the two treatments in the model.

## 3. Results

A total of 70 severe/critical COVID-19 patients were consecutively tested for HSV-1 during their hospital stay between 8 April and 31 May 2020. Of them, a total of 21 (30.0%) presented detectable HSV-1 viremia. A total of 12 patients out of 21 were also positive on the quantitative assay, with HSV-1 median viremia of 10,711 copies/mL (IQR 522-110,645). A total of 13 patients out of 21 (62%) presented HSV-1 clinical manifestations (for a total of 15 events). In particular, 2 patients developed hepatitis (9.5%), 1 of which was fulminant (4.8%), 4 pneumonia (19%), 5 herpes labialis (23.8%), 3 gingivostomatitis (14.3%), and 1 encephalitis (4.8%). Concerning treatment, 3 patients with herpes labialis (14.3%) received acyclovir 400 mg two times per day orally, while the other 10 patients (47.6%) with more severe manifestations received acyclovir 10 mg/kg three times per day intravenously.

At a later time,14 patients (5 in the HSV-1 positive, and 9 in the HSV-1 negative group) were also tested for the presence of HSV-1 in plasma at hospital admission and all were negative for HSV-1 replication.

A total of 29 patients out of 70 had a positive CMV-DNA, but only 3 with high viremia needed treatment.

Concerning SARS-CoV-2 pneumonia severity, median (IQR) PaO_2_/FiO_2_ ratio at admission was 157 mmHg (range 79–296 mmHg) in HSV-negative and 161 mmHg (range 104–187 mmHg) in HSV-positive (*p* = 0.438). A total of 23/70 (33%) patients underwent IMV, 12 (57.1%) among the HSV-positive and 11 (22.4%) among the HSV-negative (*p* = 0.005). In participants who experienced re-activation, the median time from hospital admission to the event was 19 days (IQR: 8–34). All HSV re-activations in patients who underwent IMV were diagnosed after a median time of 15 days. Table 1 describes baseline factors and biomarkers by HSV-1 status.

HSV-positive patients were more frequently treated with steroids than HSV-negative patients (76.2% vs. 49.0%, *p* = 0.036) and underwent IMV more frequently (57.1% vs. 22.4%, *p* = 0.005). Median systolic blood pressure was higher in HSV-positive patients (129 mmHg vs. 110 mmHg, *p* = 0.027). Regarding baseline biomarkers, only LDH was significantly higher in HSV-positive patients (831 vs. 609 UI/L, *p* = 0.022) (see Table 1).

The Kaplan–Meier curve (Figure 1) describes the cumulative incidence of HSV-1 viremia from 8 April 2020, when HSV screening was initiated. By day 10 after that date, the cumulative probability of experiencing HSV-1 re-activation was 29.1% (95% CI:18.0–40.3%).

On the basis of these results, universal HSV prophylaxis was started in all ventilated patients from day 19, and only two events (<140 target detected at day 26 and day 40, respectively) occurred without clinical manifestations.

Table 2A,B show the results of the logistic regression analysis investigating the association between the use of specific therapies (e.g., tocilizumab and steroids) on the risk of experiencing HSV re-activation.

In the unadjusted logistic regression analysis, use of steroid treatment was associated with an increased risk of HSV-1 re-activation (odds ratio [OR] 3.33 95%: 1.06–10.53), *p* = 0.04). Use of IMV and higher level of LDH at admission were also associated with a higher risk of HSV re-activation [25,26] (OR = 4.61 95% 1.54–13.76, *p* = 0.006, OR 16.90 95% 1.20–238.0, *p* = 0.036, respectively) while there was inconclusive evidence regarding an association with age and total lymphocyte count (OR 1.40 95% 0.92–2.14, *p* = 0.114, OR 0.90 95% 0.19–4.17, *p* = 0.892).

The association with steroid use was even stronger after controlling for previous use of both tocilizumab and IMV (OR = 5.13, 95% CI:1.36–19.32, *p* = 0.016). Interestingly, the effect size was larger when restricting to participants who were treated with a high dose of steroids, although the statistical power of this analysis was limited. In contrast, there was no association with steroid therapy duration. Moreover, there was no evidence to support an association between tocilizumab use and the risk of HSV-1 re-activation. Nevertheless, there was a signal for an exacerbation of the effect of steroids on the risk of re-activation in participants who concomitantly used tocilizumab, although the evidence for such an interaction was weak. In particular, the OR for HSV re-activation was 1.50 (95% CI 0.16–13.75) in participants who did not use tocilizumab vs. 4.15 (95% CI:1.01–17.1) for those who also used tocilizumab (*p* = 0.44).

## 4. Discussion

Our study shows that almost one-third of patients with severe/critical SARS-CoV-2 pneumonia experienced HSV-1 re-activation, with 62% of them presenting clinical manifestations, including one fulminant hepatitis.

The literature on this topic is scarce. Balc’h et al. showed that 47% (18/38) of SARS-CoV-2 pneumonia patients undergoing IMV for longer than 7 days had at least one viral pulmonary re-activation [13]. Herpesviridae re-activation was defined as two consecutive positive HSV or CMV PCR on tracheal aspirates. Nine patients had HSV re-activation, 2 CMV re-activation, and 7 had both. Patients with Herpesviridae re-activation had significantly longer duration of IMV compared to patients without. Small case series on herpetic skin lesions have been published without an evaluation of plasma PCR [27,28].

Furthermore, recently Soffritti et al. showed the presence of oral dysbiosis in COVID-19 patients compared to matched controls. Notably, oral dysbiosis correlated with symptom severity (*p* = 0.006) and increased local inflammation (*p* < 0.01). In particular, the oral virome represented 0.07% of the microbial community in controls, compared to 1.12% in COVID-19 patients. HSV-1 and EBV herpesviruses were most present [29].

Actually, in our cohort, patients presented not only plasma re-activation, but also severe clinical manifestations not previously described. Notably, two patients died of liver failure due to HSV-1 hepatitis confirmed by liver biopsy, one included in this analysis and one in a different hospital in our town [19]. Our sample size was sufficiently large to evaluate the association between type of COVID-19 therapy used and risk of HSV re-activation, after controlling for a number of identified potential confounders (PaO_2_/FiO_2_ ratio, previous use of IMV, and tocilizumab use).

It is reasonable to assume that the immunosuppression caused by SARS-CoV-2 infection, together with steroid treatment and IMV could represent a predisposing factor for herpes re-activations. In particular, before the present study a number of possible determinants of HSV-1 re-activation were identified:
The role of the virus itself. Indeed, patients with SARS-CoV-2, especially those with severe pneumonia, have a dysregulated immune response at hospitalization and often develop immune suppression characterized by lymphopenia, mainly in CD4 and CD8 T cells after the pro-inflammatory phase [30]. This virus-induced immunosuppression, followed by the administration of immunomodulatory drugs, further blocks the immune response inhibiting antiviral immunity [31]. In our study, we could test retrospectively for HSV-1 PCR at hospital admission in only 14 out of 70 patients. Despite the low number of tested patients, since they were all negative, this could suggest that SARS-CoV-2 by itself seems not to play a role in herpes re-activation.


In addition, we could not show any evidence of an association between lymphopenia and re-activation risk.
2.Tocilizumab. In our cohort, a high percentage of patients received tocilizumab with a dose higher than that used in rheumatoid arthritis [32]. In that setting, no clinically relevant manifestations of HSV-1 were described [33] and Gron et al. reported antiviral prescription in 5% of patients treated with tocilizumab without specifying the clinical reason [34]. Only one case of Herpes zoster meningitis was reported [35]. In the setting of hematological patients after CAR-T-cell infusion no statistically significant differences in risk of HSV-1 infection due to tocilizumab were reported but patients underwent herpes prophylaxis [22].

Our data do not support a role of tocilizumab on HSV re-activation in COVID-19 patients, although we hypothesized this at the time of the two fulminant hepatitis cases [19]. Actually, we found little difference in the prevalence of use of tocilizumab between HSV-positive and HSV-negative participants and after controlling for confounders, the data were highly compatible with the null hypothesis of no association. In addition, we found some evidence that the risk associated with use of steroids could be exacerbated by concomitant use of tocilizumab (>2-fold risk difference in the additive scale) but larger studies are needed to confirm this observation. In fact, even if the *p*-value is 0.44, it is possible that tocilizumab is an effect measure modifier.
3.IMV. It is well known that herpes re-activation is a common finding in patients admitted to ICU. In a pre-COVID study evaluating 201 patients with prolonged (>4 days) IMV, Luyt et al. found that 20% had HSV bronchopneumonitis with cytological and/or histological signs of deep lung infection [36]. In our analysis 57.1% of patients with HSV re-activation underwent IMV, that in the unadjusted analysis was associated with HSV-1 re-activation.4.Steroid treatment. Steroids may exacerbate herpetic re-activation of latent virus, especially among patients undergoing other stress-inducing or immunosuppressive therapies such as irradiation or chemotherapy [37]. Case reports have been described in patients with inflammatory bowel disease [38]. Our analysis shows convincing evidence for an association between use of steroids and risk of HSV-1 re-activation. The association was even stronger after controlling for previous use of both tocilizumab and IMV (OR = 5.13, *p* = 0.016). Interestingly, the effect size was larger when restricting to participants who were treated with high-dose steroids, while there was no association with duration of steroid treatment.

This evidence is even more relevant since after the period described in the study, the RECOVERY trial showed that dexamethasone decreases mortality in patients with oxygen need and it is now considered the SOC in patients hospitalized for SARS-CoV-2 pneumonia.

Our observations had an important clinical impact in response to the first wave of COVID-19 pandemic as they prompted around the end of April 2020 the initiation of acyclovir prophylaxis in all patients with SARS-CoV-2 pneumonia either admitted to ICU or in non-invasive ventilation. In all the other patients, we continued to perform HSV-1 plasma PCR twice a week and we started prophylaxis or treatment according to test results. After the implementation of acyclovir prophylaxis, we observed no new clinical manifestations due to HSV-1 re-activation.

Luyt et al. recently published a randomized clinical trial in 238 adults demonstrating that, in patients receiving mechanical ventilation for 96 h or more with HSV re-activation in the throat, use of acyclovir, 5 mg/kg, 3 times daily for 14 days, did not increase the number of ventilator-free days at day 60, compared with placebo [39]. The observed difference for mortality was nonsignificant but may be worthy of further investigation in subsequent more highly powered studies.

Our study has some limitations: first of all, it is a single-center study enrolling severely to critically ill patients, thus, results cannot be generalized to all COVID-19 patients; second, screening for HSV-1 was not performed routinely in all admitted patients as it was prompted by a serious adverse event. However, since participant inclusion was systematic on the basis of calendar date, this is unlikely to have introduced important selection bias. Due to this limitation, we have HSV-PCR at admission only in 14 patients out of 70. Finally, because of the observational nature of the study, we cannot prove that steroid use was a cause of the episodes of HSV-1 re-activations observed.

Our study has some strengths: it is the first study that analyzed the incidence and clinical implications of HSV-1 re-activation in patients with SARS-CoV-2 pneumonia; second it has strong clinical and therapeutic implications for COVID-19 patients, especially in the present and future waves of hospitalized patients most of whom are treated with steroids, which is now considered the SOC.

In conclusion, our study shows a high incidence of both virological and clinical HSV-1 re-activation in patients with SARS-CoV-2 severe/critical pneumonia. Data show an association between this risk and treatment with steroids, which could not be explained by age, previous IMV, and level of inflammation at hospital admission. Further studies are needed, especially a randomized controlled trial, to confirm the utility of acyclovir prophylaxis in COVID-19 patients with severe pneumonia admitted to the hospital.

## Figures and Tables

**Figure 1 microorganisms-09-01896-f001:**
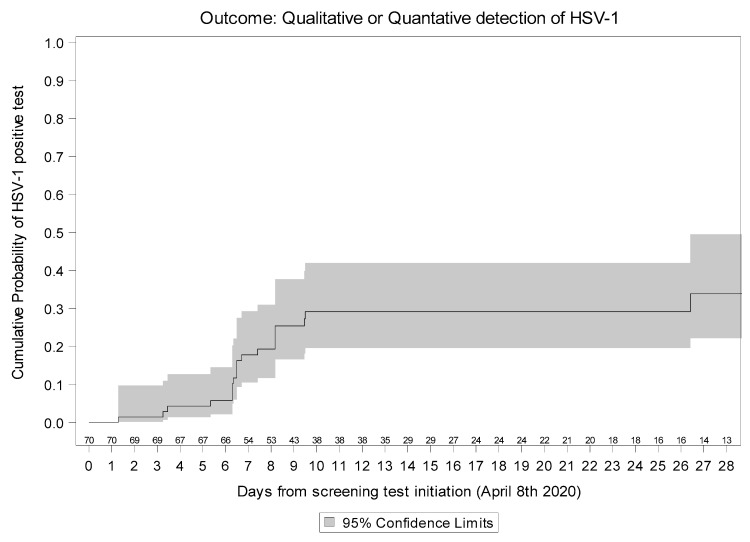
Kaplan–Meier estimates of HSV re-activation.

**Table 1 microorganisms-09-01896-t001:** Baseline characteristics, comorbidities and outcome of COVID-19 patients by HSV-1 status.

	HSV−1			
	Positive	Negative	*p*-Value *	Total
	N = 21	N = 49		N = 70
** *Patients’ Characteristics* **				
**Age,** years Median (IQR)	72 (66, 76)	67 (52, 76)	0.185	70 (58, 76)
**BMI,** Kg/m^2^ (49)	27.5 (25.6, 32.8)	26.2 (24.2, 29.9)	0.150	26.7 (24.6, 31.1)
** *Any comorbidity, n (%)* **				
Yes	15 (71.4%)	28 (57.1%)	0.264	43 (61.4%)
** *Comorbidities, n (%)* **				
Diabetes	6 (28.6%)	8 (16.3%)	0.244	14 (20.0%)
Hypertension	13 (61.9%)	25 (51.0%)	0.406	38 (54.3%)
Cardiovascular Disease	3 (14.3%)	10 (20.4%)	0.549	13 (18.6%)
Chronic Kidney Disease	2 (9.5%)	4 (8.2%)	0.853	6 (8.6%)
Cancer	0 (0.0%)	3 (6.1%)	0.250	3 (4.3%)
Hepatitis B/C	0 (0.0%)	0 (0.0%)		0 (0.0%)
** *Signs and symptoms, n (%)* **				
Fever, median (IQR)	36 (36, 36)	36 (36, 37)	0.521	36 (36, 36)
Cough	3 (14.3%)	14 (28.6%)	0.205	17 (24.3%)
Myalgia	0 (0.0%)	4 (8.2%)	0.181	4 (5.7%)
Sputum	1 (4.8%)	1 (2.0%)	0.534	2 (2.9%)
Headache	1 (4.8%)	3 (6.1%)	0.823	4 (5.7%)
Haemoptysis	0 (0.0%)	0 (0.0%)		0 (0.0%)
Diarrhea	0 (0.0%)	0 (0.0%)		0 (0.0%)
Systolic pressure, mmHg median (IQR)	129 (120, 140)	110 (101, 130)	0.027	120 (110, 135)
Respiratory rate, % median (IQR)	22 (20, 36)	22 (20, 27)	0.490	22 (20, 30)
** *Baseline* ** **PaO_2_/FiO_2_**	161 (104, 187)	157 (79, 296)	0.438	159 (80, 285)
** *SOFA Score* **	2 (0, 4)	2 (0, 4)	0.730	2 (0, 4)
** *Markers, Median (IQR)* **				
Haemoglobin, g/L	12.6 (10.3, 13.9)	12.4 (11.4, 13.5)	0.934	12.5 (10.8, 13.8)
White cells, mm^3^	6510 (5170, 8490)	6180 (5190, 8200)	0.729	6365 (5170, 8490)
Total lymphocytes, N	1791 (570.0, 2519)	1290 (810.0, 2383)	0.984	1358 (700.0, 2519)
Total lymphocytes, %	27.9 (7.8, 30.9)	20.3 (8.8, 36.0)	0.625	22.4 (8.6, 33.9)
Alanine amino-transferase, U/L	39.0 (29.0, 69.0)	48.0 (31.0, 81.0)	0.513	41.5 (29.0, 81.0)
Bilirubin, mg/L	0.6 (0.4, 0.9)	0.5 (0.4, 0.8)	0.366	0.6 (0.4, 0.8)
Calcium, mg/L	8.4 (8.2, 8.7)	8.6 (8.2, 8.9)	0.363	8.5 (8.2, 8.9)
Creatine Kinase, U/L	127.0 (64.0, 305.0)	78.0 (33.0, 180.0)	0.106	97.5 (36.0, 206.0)
Chloride, mmol/L	100.5 (98.0, 104.0)	100.0 (96.0, 103.0)	0.454	100.0 (97.0, 104.0)
Creatinine, mg/L	0.8 (0.7, 1.0)	0.9 (0.7, 1.1)	0.353	0.8 (0.7, 1.1)
D-dimer, mg/L	1200 (810.0, 2650)	900.0 (460.0, 1800)	0.088	1060 (580.0, 2070)
Lactate dehydrogenase, U/L	831.0 (556.0, 998.0)	609.0 (466.0, 745.0)	0.022	652.0 (473.0, 832.0)
C-reactive protein, mg/L	15.5 (5.0, 22.2)	7.5 (4.5, 18.9)	0.405	9.0 (4.5, 19.7)
Platelets, 109/L	182.0 (140.0, 244.0)	180.0 (151.0, 251.0)	0.888	181.0 (149.0, 251.0)
Potassium, mmol/L	3.7 (3.5, 3.9)	3.8 (3.6, 4.0)	0.213	3.7 (3.5, 4.0)
Sodium, mmol/L	137.0 (134.0, 139.0)	136.0 (135.0, 139.0)	0.663	136.0 (135.0, 139.0)
IL−6, mg/L	412.8 (241.1, 1252)	253.4 (78.9, 1418)	0.392	280.5 (92.8, 1349)
Ferritin, mg/L	987.5 (472.5, 1475)	603.5 (416.0, 1562)	0.684	688.0 (423.0, 1518)
** *Disease Duration* **				
Days from symptoms onset to hospitalisation, median (IQR)	5 (2, 7)	8 (4, 15)	0.572	7 (3, 12)
Days from hospitalisation to intubation, median (IQR)	4 (2, 7)	4 (1, 6)	0.827	4 (2, 6)
** *Follow-up, days* **	7 (3, 24)	14 (6, 27)	0.170	13 (6, 25)
** *Intervention, n (%)* **			0.027	
Tocilizumab subcutaneous	3 (27.3%)	5 (12.5%)		8 (15.7%)
Tocilizumab intravenous	11 (52.4%)	30 (61.2%)		41 (58.6%)
Only SOC	7 (33.3%)	13 (26.5%)		20 (28.6%)
Steroids	16 (76.2%)	24 (49.0%)	0.036	40 (57.1%)
** *Outcomes* **				
** *Events, n (%)* **				
Invasive mechanical ventilation	12 (57.1%)	11 (22.4%)	0.005	23 (32.9%)
Death-all	6 (28.6%)	9 (18.4%)	0.344	15 (21.4%)

* Chi-square or Mann–Whitney U or Kruskal–Wallis test as appropriate. HSV: herpes simplex virus; N: number; IQR: interquartile range, BMI: body mass index; PaO_2_/FiO_2_: ratio of arterial oxygen partial pressure (PaO_2_) to fractional inspired oxygen (FiO_2_); SOC: standard of care.

**Table 2 microorganisms-09-01896-t002:** Logistic regression estimates of the risk of HSV-1 infection.

(A)				
	Unadjusted		Adjusted *	
	Odds Ratio (95% CI)	*p*-Value	Odds Ratio (95% CI)	*p*-Value
** *Steroids* **				
Yes (any dose) vs. No	3.33 (1.06, 10.53)	0.040	5.13 (1.36, 19.32)	0.016
Low dose vs. No	3.06 (0.90, 10.33)	0.072	4.80 (1.20, 19.26)	0.027
High dose vs. No	4.17 (0.91, 19.18)	0.067	6.16 (1.06, 35.74)	0.043
per day longer exposure	1.04 (0.89, 1.22)	0.625	1.07 (0.90, 1.26)	0.461
**(B)**				
	**Unadjusted**		**Adjusted ***	
	**Odds Ratio (95% CI)**	***p*-Value**	**Odds Ratio (95% CI)**	***p*-Value**
** *Use of tocilizumab* **				
Yes vs. No	1.87 (0.54, 6.53)	0.323	1.91 (0.36, 10.21)	0.452

(**A**) * adjusted for previous use of tocilizumab and invasive mechanical ventilation. (**B**) * adjusted for PaO_2_/FiO_2_ ratio. HSV: herpes simplex virus; PaO_2_/FiO_2_: ratio of arterial oxygen partial pressure (PaO_2_) to fractional inspired oxygen (FiO_2_).

## Data Availability

Data available on request due to privacy restrictions.

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
