# Peer review of "Herpes Simplex Virus Re-Activation in Patients with SARS-CoV-2 Pneumonia: A Prospective, Observational Study"

_microorganisms, 2021, doi:10.3390/microorganisms9091896_

Round 1

Reviewer 1 Report

The pandemic of COVID-19 has claimed millions of lives globally. 
With the number of COVID-19 deaths on the rise around the world, approaches to reduce the spread of the virus are of crucial importance.
This study, with the aim to evaluate the herpetic reactivations in 70 hospitalized patients affected by severe pneumonia caused by SARS-CoV-2, showed a significant incidence of herpetic reactivation. Overall, the manuscript is well written, the experiments are well designed and the data a presented appropriately, but some issues listed below need to be addressed.

- In the title modify "SARSCOV-2" with "SARS-CoV-2"
-In line 78 they speak about "previous studies". Can add a reference?
- Please rectify  "ml" in "mL" in Line number 99, 123. 
- In line 79 modify "Herpesviridae" in italics
- In line 81 the authors said "Regarding patients in non-ICU wards, only case reports on the skin, oral and ocular Herpesviridae clinical manifestations were published". Please cite also publication on the  Epstein-Barr Virus Seroprevalence  ( DOI: 10.1159/000496828) 
- In the introduction, it is also important to focalize attention to ocular infections widespread in hospitals.

Reviewer 2 Report

This study analysed HSV-1 reactivation in SARS-CoV-2 patients admitted to ICU and the associated risk factors that could lead to reactivation. The study showed interesting results that high LDH, IMV and steroid use may contribute to reactivation of HSV-1 and these results have led to prophylaxis being administered to high risk SARS-CoV-2 patients. Issues with the study are mentioned below:

Lines 73 – 81 need to be reworded. The initial sentence is not relevant to the study as it cites studies looking at co-infections between SARS-CoV-2 and influenza, rather than HSV. Lines 75 – 78 are particularly confusing because it reads as if these cited references were performed on ICU patients without SARS-CoV-2. Needs to be made clearer that in ICU patients, there are higher rates of HSV reactivation in those positive for SARS-CoV-2 compared to those negative.

PCR methods are vague and much of the detail is missing. What sample types were the initial qualitative PCR experiments performed on? The PCR kits used (HSV-1 ELITe MGB) are described as qualitative assays on the manufacturer’s website, but this study has used them to quantify virus levels. How was this kit modified to accommodate this?

Was any QC performed on the samples prior to quantification to check for RNA integrity? This is especially important given that the kit was originally designed for qualitative purposes. Any compromise to the RNA integrity could affect the qualitative data.

Only 14 out the 70 total patients (5 out of the total 21 HSV positive) were tested for HSV reactivation prior to hospital admission. It is unknown whether the remaining HSV positive participants were positive prior to admission or due to treatment in ICU. Given that HSV reactivation is common, and even more so in COVID patients, this is a major control missing from the data set that needs to be addressed.

The results do not mention that systolic pressure is significantly higher in HSV positive patients vs HSV negative (p = 0.027). This needs to be addressed and controlled for.

Discussion line 272 – 274 claims that SARS-CoV-2 by itself did not appear to be a risk factor for herpes reactivation since all our patients had a negative HSV-1 PCR at hospital admission. This is not true as you previously stated you later went back to check for reactivation at hospital admission but only tested 14 out the 70 patients.

Round 2

Reviewer 2 Report

Concerns have been addressed and changes made to the manuscript to address these concerns and limitations.